# Crosswalk between Charlson Comorbidity Index and the American Society of Anesthesiologists Physical Status Score for Geriatric Trauma Assessment

**DOI:** 10.3390/healthcare11081137

**Published:** 2023-04-15

**Authors:** Oluwaseun John Adeyemi, Ariana Meltzer-Bruhn, Garrett Esper, Charles DiMaggio, Corita Grudzen, Joshua Chodosh, Sanjit Konda

**Affiliations:** 1Ronald O. Perelman Department of Emergency Medicine, New York University Grossman School of Medicine, New York, NY 10016, USA; 2Department of Orthopedic Surgery, New York University Grossman School of Medicine, New York, NY 10016, USA; ariana.meltzer-bruhn@nyulangone.org (A.M.-B.); garrett.esper@nyulangone.org (G.E.); sanjit.konda@nyulangone.org (S.K.); 3Department of Surgery, New York University Grossman School of Medicine, New York, NY 10016, USA; charles.dimaggio@nyulangone.org; 4Department of Population Health, New York University Grossman School of Medicine, New York, NY 10016, USA; joshua.chodosh@nyulangone.org; 5Department of Medicine, Memorial Sloan Kettering Cancer Center, West Harrison, NY 10604, USA; grudzenc@mskcc.org; 6Department of Medicine, New York University Grossman School of Medicine, New York, NY 10016, USA

**Keywords:** American Society of Anesthesiologists Physical Status, Charlson Comorbidity Index, geriatric trauma, area under the curve, partially proportional ordinal regression, crosswalk, trauma models, receiver operating characteristics, surgery

## Abstract

The American Society of Anesthesiologists Physical Status (ASA-PS) grade better risk stratifies geriatric trauma patients, but it is only reported in patients scheduled for surgery. The Charlson Comorbidity Index (CCI), however, is available for all patients. This study aims to create a crosswalk from the CCI to ASA-PS. Geriatric trauma cases, aged 55 years and older with both ASA-PS and CCI values (N = 4223), were used for the analysis. We assessed the relationship between CCI and ASA-PS, adjusting for age, sex, marital status, and body mass index. We reported the predicted probabilities and the receiver operating characteristics. A CCI of zero was highly predictive of ASA-PS grade 1 or 2, and a CCI of 1 or higher was highly predictive of ASA-PS grade 3 or 4. Additionally, while a CCI of 3 predicted ASA-PS grade 4, a CCI of 4 and higher exhibited greater accuracy in predicting ASA-PS grade 4. We created a formula that may accurately situate a geriatric trauma patient in the appropriate ASA-PS grade after adjusting for age, sex, marital status, and body mass index. In conclusion, ASA-PS grades can be predicted from CCI, and this may aid in generating more predictive trauma models.

## 1. Introduction

Geriatric trauma is the fifth leading cause of mortality among older adults in the U.S. and accounts for a quarter of inpatient geriatric admissions nationally [1]. These injuries commonly occur from falls, motor-vehicle crashes, and burn injuries [2], and approximately 80 percent of geriatric trauma cases are managed non-surgically [3]. Geriatric trauma patients, aged 55 years and older [4], are a unique trauma population that is at increased mortality risk compared to trauma patients less than 55 years old [5,6,7,8,9]. Multiple factors have been associated with increased geriatric trauma mortality risk, some of which include frailty [10,11], cognitive impairment [1], cardiovascular and pulmonary insufficiency [1], and poor injury triage [6,12,13]. With the aging U.S. population [14], it is expected that geriatric trauma cases will increase over time. Improving injury outcomes among geriatric trauma patients, therefore, has been a priority area for quality assessment and improvement by the American College of Surgeons Trauma Quality Improvement Program [15].

The American College of Surgeons Trauma Quality Improvement Program emphasizes quality improvements across hospitals in the United States [16]. To accomplish this, the program collects data from individual trauma centers to provide feedback on performance and patient outcomes and identifies potential institutional changes that can be implemented to improve care quality and patient outcomes [16]. These quality assessments help standardize care for all patient populations, especially geriatric trauma patients. To assess care quality, trauma centers retrospectively evaluate expected and actual outcomes with the American Society of Anesthesiologists Physical Status (ASA-PS) [17,18,19] and the Charlson Comorbidity Index (CCI) [17,20,21], being the more common comorbidity measures being used.

The ASA-PS is a six-item grade, typically assigned by an anesthesia professional, for patients scheduled for surgery. While grades 5 and 6 identify moribund and dying patients, patients assigned the remaining four ASA-PS grades are sometimes categorized as low-risk (grades 1 and 2) and high-risk (grades 3 and 4) [22,23]. The CCI, a measure of patients’ mortality risks, assigns values ranging from one to six to distinct comorbidities [24,25]. These comorbid conditions include myocardial infarction, congestive cardiac failure, peripheral vascular disease, cerebrovascular accident or transient ischemic attack, dementia, chronic obstructive pulmonary disease, connective tissue disease, peptic ulcer disease, liver disease, diabetes, hemiplegia, chronic kidney disease, metastatic and non-metastatic solid tumor, leukemia, lymphoma, and acquired immune deficiency syndrome [24,25]. The CCI has been used for risk triaging, care protocol formulation, and informing patient management [25,26,27]. Additionally, the CCI can be computed for trauma and non-trauma patients. However, earlier studies have shown the superior predictive ability of the ASA-PS over the CCI [28,29].

To standardize quality comparisons between surgical and non-surgical trauma patients during retrospective quality “look-backs” (as is routinely done in trauma centers’ quality improvement programs mandated by the American College of Surgeons), a method of assigning non-surgical patients an ASA-PS grade retrospectively is needed. Providing a crosswalk that identifies a predicted ASA grade for all geriatric trauma patients will provide a uniform measure of clinical assessment without excluding the larger proportion of geriatric trauma patients that are managed non-surgically. The aim of this study, therefore, is to create a crosswalk between CCI and ASA-PS by predicting ASA-PS grades using CCI values while evaluating the accuracy of such predictions among geriatric trauma patients. We hypothesized that the CCI would exhibit acceptable levels of accuracy in predicting ASA-PS grades among geriatric trauma patients. 

## 2. Materials and Methods

### 2.1. Study Design, Setting, and Patients

For this retrospective cohort study, we pooled a single institution’s trauma data between 2014 and 2020 from the electronic health record of an urban academic hospital with level-I trauma center designation in New York. This study was part of the validation studies for the implementation of a novel scoring tool for geriatric trauma triage (Institutional Review Board (IRB): s15-00371), and the protocol of the investigation has been published earlier [30,31,32,33]. All relevant de-identified patient data were recorded in an IRB-approved database. 

### 2.2. Inclusion and Exclusion Criteria

The inclusion criteria consisted of adult patients aged 55 years and older, who were admitted through the emergency department, and who received orthopedic trauma care between 2014 and 2020 (N = 12,303) (Figure 1). We used age 55 as the cutoff since injury-related mortality significantly increases from age 55 years and higher [4]. We restricted the data to cases that had an ASA-PS grade (n = 4229) and excluded cases that were ASA-PS grade 5 (n = 11). In this study, there were no patients with ASA-PS grade 6, and we excluded patients with ASA-PS grade 5 (n = 11) due to the small counts. Additionally, we excluded cases without reported CCI values (n = 6). The final dataset (n = 4223) was split, using a simple random sampling technique, in a 70:30 ratio into the training dataset (n = 2956) and the test dataset (n = 1267). The training dataset was used for modeling the relationship between ASA-PS and CCI, while the test dataset was used for internal validation.

### 2.3. Variable Definitions

The two primary variables for this analysis were the ASA-PS grade—the outcome variable—and the CCI values—the predictor variable. ASA-PS was defined in two ways: as an ordinal value ranging from 1 to 4 (having excluded higher ASA-PS grades) and as a binary variable (<3 (low risk) and ≥3 (high risk)). We defined CCI in three forms: (1) as a five-point categorical variable ranging from 0 (no comorbidity) to 1, 2, 3, and 4+, representing one, two, three, and four or more reported comorbidities; (2) as a binary variable (0 and 1 or higher); and (3) as a continuous variable. These CCI definitions represent the different ways CCI has been used in clinical research [22,34,35,36,37]. 

We selected age, sex, marital status, and body mass index (BMI) as additional control variables. Age was measured as a continuous variable, while sex was measured as a binary variable. Marital status was defined as a four-point categorical variable—single, married, divorced, and widowed. BMI was defined as a four-point categorical variable—underweight (<18.5 kg/m^2^), normal weight (18.5 to 24.9 kg/m^2^), overweight (25.0 to 29.9 kg/m^2^), and obese (≥30 kg/m^2^).

### 2.4. Data Analysis

Before dividing the data into the training and test datasets, we assessed for missingness in the control variables. Missing values were observed in the sex (5%), marital status (9%), and BMI (31%) variables. We assessed the pattern of missingness, which showed that the missingness was not “missing completely at random” [38], evidenced by the significant Little’s test (*p* < 0.001) [39]. We imputed the missing values using the multivariate imputation by chained equation (MICE) [40]. Since higher values of imputation yield better estimates and a minimum of 10 iterations have been recommended [41,42], we selected 31 to correspond to the maximum proportion of missing values in this study. Thereafter, we reported the counts and proportions for the categorical variables and the mean and standard deviation for the continuous variables.

### 2.5. Model Testing and Internal Validation

Using the ASA-PS binary models, we performed a receiver operating characteristic (ROC) analysis to assess the discriminative property of CCI-based predictive probabilities on ASA-PS. Our ROC analysis was performed in two steps: (1) univariate logistic regression and (2) computation of areas under of curve from predicted probabilities of the logistic regression analysis. For each set of univariate analysis, the five outcome measures were the four dummy definitions of the ASA-PS (for example, ASA-PS 1: Yes or No, ASA-PS 2: Yes or No, etc.) and the high-risk/low-risk ASA-PS binary categorization. For each of these five binary outcome variables, the predictors were the five dummy variable definitions of CCI (for example, CCI 0: Yes or No, CCI 1: Yes or No, etc.). Essentially, we had 25 univariate logistic regression models, i.e., each of the five ASA-PS binary variables had five CCI dummy variable predictors. We performed the univariate logistic regression analyses using the training dataset and reported the unadjusted odds ratios for all 25 models and computed the predicted probabilities for each patient in the models that had significant associations. We validated the regression models by repeating the same regression analyses using the test dataset. We generated predicted probabilities for each patient in the models that showed significant associations. Of the 25 regression analyses, eight models showed significant associations between ASA-PS and CCI. Thereafter, we performed ROC analyses across the eight significant models using the training and test datasets. For each ROC analysis, the ASA-PS binary variable was the reference variable, and its associated predicted probability variable was the classification variable. We reported the pseudo-r-squared values, the sensitivity, specificity, and the area under the curve of each ROC analysis (AUROC). We interpreted the AUROC ranging from 0.7 to 0.8 as acceptable, 0.8 to 0.9 as excellent, and values higher than 0.9 as outstanding [43]. Additionally, we computed Youden’s index, a measure of the model’s effectiveness as a diagnostic marker [44]. Values of 0.5 or higher were deemed adequate [44]. For ASA-PS models that had more than one significant CCI predictor, we assessed the differences in the AUROC using DeLong’s test [45].

### 2.6. ASA-PS and CCI Crosswalk

Our first step in creating a crosswalk between ASA-PS and CCI was to select the variables in the model. The main predictor was CCI, measured as a categorical variable. We selected covariates that are conventional demographic characteristics, similar to prior studies [22,33]. We limited our covariate selection to variables that are routinely entered in the electronic health records (such as age, sex, marital status, and body mass index) and excluded variables that are typically associated with high rates of underreporting (such as smoking history) or non-response (such as race/ethnicity). Using the training dataset, we reported the association between the binary categorizations of ASA-PS and the covariates. Specifically, we performed a covariate forward stepwise selection of the least absolute shrinkage and selection operator (LASSO) binary logistic regression analysis to select the variables for the crosswalk [46,47]. Using the full dataset, we performed a multivariate partially proportional ordinal logistic regression using the ASA-PS as a four-point ordinal variable. The choice of a partially proportional ordinal logistic regression was the failed parallel test assumption, evident by the significant Brant test [48]. We generated point estimates from the ordinal-regression-model equation and created a formula to predict each ASA-PS grade from CCI. Data were analyzed with STATA version 17 for Windows (StataCorp, College Station, TX, USA) [49].

## 3. Results

### 3.1. Population Characteristics

We pooled 2956 and 1267 adults, aged 55 years and older, in the training and test datasets, respectively (Table 1). In the training dataset, the mean (SD) age was 76.7 (12.7) years, and the sample population was mostly females (71%), married (42.5%), and with normal BMI (41.4%). Approximately 43% of the population in the training dataset had a CCI of zero. Half of the training dataset had an ASA-PS grade of 3, and approximately 64% of the training-dataset population had ASA-PS grades of 3 or 4. The demographic characteristics and the distributions of the CCI and ASA-PS categories in the test dataset were largely similar to the training dataset. In the training dataset, 88% and 67% of the patients with ASA-PS grades 1 and 2, respectively, had a CCI value of zero (Table 2). Among the patients with ASA-PS grades 3 and 4, the predominant CCI value was two. CCI values of four or higher produced a zero count under ASA-PS grade 1. Contrastingly, 90% of patients with a CCI of four or higher had ASA-PS grades 3 or 4.

### 3.2. Univariate Regression Analysis

A unit increase in age was significantly associated with decreased odds of being an ASA-PS grade 1 or 2 and significantly associated with increased odds of being an ASA-PS grade 3 or 4 (Table 3). Female patients were more likely to have an ASA-PS grade of 2 or less and less likely to have an ASA-PS grade of 3 or 4. Compared to single patients, patients who were widowed were significantly less likely to have ASA-PS grades 1 or 2 but significantly more likely to have ASA-PS grades 3 or 4. CCI values of 1 and 2 were significantly less likely to be ASA-PS grades 1 or 2 but significantly more likely to be ASA-PS grades 3 or 4. Similarly, CCI values of 3 or 4 and higher were significantly less likely to be an ASA-PS grade 2 but significantly more likely to be ASA-PS grade 3 or 4. A unit increase in CCI was less likely to be ASA-PS grades 1 or 2 but more likely to be ASA-PS grades 3 or 4.

When ASA-PS was considered a binary variable (high- and low-risk surgical groups), a unit increase in age was associated with 66% (odds ratio (OR): 1.66; 95% CI: 1.05–1.07) increased odds of being an ASA-PS grade 3 or 4. Female patients were 28% significantly less likely to be an ASA-PS grade 3 or 4 (OR: 0.72; 95% CI: 0.61–0.85). Compared to single patients, patients who were married were 20% less likely to have ASA-PS grades 3 or 4 (OR: 0.80; 95% CI: 0.66–0.96), while those who were widowed were 151% more likely to have an ASA-PS of 3 or 4 (OR: 2.51; 95% CI: 1.97–3.21). CCI value of 1 had 3.7 times the odds of being an ASA-PS grade 3 or 4 (95% CI: 3.08–4.55), while a CCI value of 4 or higher had 11.8 times the odds of being an ASA-PS grade 3 or 4 (95% CI: 7.89–11.75). A unit increase in CCI was associated with 1.94 times the odds of being an ASA-PS grade 3 or 4 (95% CI: 1.79–2.10).

### 3.3. Multivariate Binary Logistic Regression: Prediction of ASA-PS Categories

Using the LASSO selection technique, four of the five multivariate models (ASA-PS 2, 3, 4 and ASA-PS Binary) included age, sex, marital status, and BMI as control variables. The LASSO selection excluded marital status from the ASA-PS grade 1 model. A CCI value of zero was most predictive of ASA-PS grades 1 and 2 with adjusted odds ratios (AOR) of 6.83 (95% CI: 3.36–13.90) and 3.75 (95% CI: 3.16–4.45), respectively (Figure 2). CCI values of 1 (AOR: 1.67; 95% CI: 1.41–1.98) and 2 (AOR: 1.64; 95% CI: 1.31–2.05) were most predictive of ASA-PS grade 3. CCI values of 2, 3, and 4 were predictive of ASA-PS grade 4, and the adjusted odds ratio increased in a dose–response pattern from 1.65 (95% CI: 1.25–2.17) to 2.59 (95% CI: 1.88–3.58) to 4.00 (95% CI: 3.02–5.29). A CCI of 1 or higher was predictive of a high-risk surgical group (ASA-PS 3 or 4) with an AOR of 4.59 (95% CI: 3.85–5.86).

### 3.4. Discriminant Properties

In the training dataset, a CCI value of zero exhibited 85% (95% CI: 80.4–90.0; Youden’s index = 0.62) accuracy in predicting ASA-PS grade 1 and 77% (95% CI: 74.7–88.4; Youden’s index = 0.43) accuracy in predicting ASA-PS grade 2 (Table 4). Both CCI values of 1 and 2 significantly predicted ASA-PS grade 3. However, the accuracy was below acceptable levels, and neither CCI value of 1 or 2 exhibited superior predictive ability above the other (DeLong’s test *p*-value = 0.587 (not significant)). CCI values of 2, 3, and 4 or higher significantly predicted ASA-PS grade 4 at 69% (95% CI: 66.0–71.3; Youden’s index = 0.28), 70% (95% CI: 67.7–72.9; Youden’s index = 0.30), and 73% (95% CI: 70.1 to 75.1; Youden’s index = 0.35) accuracies, respectively. However, the CCI of 4 or higher exhibited the most superior discriminant quality evidenced by the significant DeLong’s test when compared to models with CCI values of 2 (*p*-value < 0.001) and 3 (*p*-value = 0.047). Furthermore, CCI values of 1 or higher significantly predicted ASA-PS 3 or 4 (high-risk surgical group) 79.5% of the time with an acceptable level of accuracy (95% CI: 77.7–81.2; Youden’s index = 0.46).

In the test dataset, a CCI value of zero excellently predicted ASA-PS grade 1 with an AUROC of 89% (95% CI: 85.1–93.1) (Figure 3). A CCI value of 0 accurately predicted ASA-PS grade 2 within acceptable limits (AUROC: 76.5%; 95% CI: 73.7–79.3), and ASA-PS grade 4 was accurately predicted by CCI values of 2 (AUROC: 74.8%; 95% CI: 70.9–78.7), 3 (AUROC: 75.2%; 95% CI: 71.4–79.1), and 4 (AUROC: 76.2%; 95% CI: 72.4–80.0). Additionally, CCI values of 1 and higher excellently predicted high-risk surgical groups (ASA-PS 3 and 4) (AUROC: 80.2%; 95% CI: 77.6–82.7).

### 3.5. Multivariate Ordinal Logistic Regression: Prediction Probabilities

Using the multivariate ordinal logistic model, we created a predictive model to compute the predictive probabilities of each ASA-PS grade. Using the formula created from the partially proportional ordinal logistic regression model, the ASA-PS grades 1, 2, 3, and 4 can be calculated (Table 5). The grade with the highest predicted probability is the ASA-PS grade of the patient. For example, this formula can be used to assess the predicted ASA-PS of two hypothetical 62-year-old married men, both overweight, but one (patient X) having no comorbid illness (CCI = 0) and the other (patient Y) having a CCI of 3. The *y*-values for patients X and Y are 6.135 and 8.121, respectively. Patient X’s predicted probability percentages for ASA-PS 1, 2, 3, and 4 are 4.2%, 73.5%, 22.2%, and 0.1%, respectively. Patient Y’s predicted probability percentages for ASA-PS 1, 2, 3, and 4 are 0.6%, 31.8%, 67.0%, and 0.6%, respectively. Based on these values, patients X and Y will have predicted ASA-PS grades 2 and 3, respectively.

## 4. Discussion

In this study, we assessed the predictiveness of CCI in identifying each ASA-PS grade as well as high- and low-risk ASA-PS categories. We report that a CCI of zero is highly predictive of ASA-PS grade 1 or 2, while a CCI of 1 or higher is highly predictive of ASA-PS grade 3 or 4. While a CCI of 1 solely predicts ASA-PS grade 3, a CCI of 2 may predict ASA-PS grade 3 or 4. Additionally, while a CCI of 3 solely predicts ASA-PS grade 4, a CCI of 4 and higher exhibited greater accuracy in predicting ASA-PS grade 4. We further present a formula that may accurately situate a geriatric trauma patient in the appropriate ASA-PS grade after adjusting for age, sex, marital status, and body mass index.

Mannion and colleagues [22] were the first to predict ASA-PS grades from CCI, albeit among spine patients scheduled for surgery. We extended this novel field of knowledge to include all geriatric trauma patients. Indeed, it could be cumbersome to engage in lengthy statistical computations if the goal of assessing the predicted ASA-PS is to generate a “rough estimate” of the ASA-PS categories a non-surgical trauma patient will be assigned. This study addresses such concerns, and we reported that a CCI of 1 or higher accurately predicts high-risk ASA-PS categories, similar to the crosswalk created by Mannion et al. In addition to providing a “rough estimate”, we provided a statistical crosswalk between CCI and ASA-PS while accounting for age, sex, marital status, and body mass index. 

Many studies have reported improvement in the predictive models when ASA-PS is added to the model specifications [29,33,50]. Ringdal et al. [50] reported that the ASA-PS scale can reliably classify comorbidities among trauma patients, Konda et al. [33] reported that ASA-PS improved the predictive ability of a validated geriatric trauma triage tool, and Tran et al. [29] reported that pre-injury ASA-PS is an independent predictor of re-admission after a major traumatic injury. The versatility of the ASA-PS, especially across the clinical trajectory of geriatric trauma patients, makes it an important metric in trauma quality improvement studies. However, with ASA-PS restricted to patients who underwent surgery, a substantial proportion of eligible cases, which would otherwise have strengthened the model, are excluded. This study fills that void by providing the toolkit for generating predictable ASA-PS estimates with acceptable levels of accuracy.

Indeed, neither the CCI nor the ASA-PS is without weaknesses, but this crosswalk provides an opportunity for continued research to improve geriatric trauma outcomes. The strengths of the CCI, for example, include its relative ease of calculation and its availability for both surgical and non-surgical geriatric trauma patients. However, the CCI is limited to 16 chronic conditions [24], and since its development in 1987, the weights allocated to each illness category have become obsolete due to advancements in clinical care [26]. Additionally, three decades after its creation, more chronic conditions have been identified as significant predictors of mortality among older adults, such as valvular heart disease, cardiac arrhythmias, and alcohol and drug abuse [26,51]. We circumvented this limitation by calculating the index of diseases and using age as a separate predictor, consistent with Mannion et al.’s seminal paper [22]. The ASA-PS, while being a more predictive measure of morbidity and mortality [28,29], is limited to patients scheduled for surgery. Additionally, the ASA-PS is officially assigned by an anesthesiologist on the day of the surgery [52], and has a weak inter-rater agreement [22,53]. This crosswalk provides a more objective measure of ASA-PS allocation for pre-operative care, and early assignment of an ASA-PS grade may inform the need for geriatric consult activation or multi-disciplinary specialty care. Furthermore, the crosswalk provides an opportunity for researchers to extend the frontiers of geriatric trauma care by creating predictive tools that will improve injury triage.

This study has its limitations. The study design is cross-sectional, and causal inferences cannot be made. Additionally, we pooled data from a single center, and our data is limited to adults aged 55 years and older. Hence, our study has limited generalizability. Some of the models had low adjusted R-squared values. While this finding is a concern, other authors have reported a similar pattern when predicting ASA-PS [22,54]. The addition of more demographic or injury characteristics may improve the adjusted R-squared values of such models but may undermine the usability of the crosswalk if such variables are not captured retrospectively in the electronic health records. Regardless of the low R-squared values, the acceptable level of accuracy of most of the models can guide researchers interested in the CCI to ASA-PS crosswalk. Despite these limitations, this study provides a means of computing ASA-PS grades for non-surgical geriatric trauma patients to aid retrospective quality “look-backs” without excluding non-surgical patients due to the absence of ASA-PS grades. Additionally, this crosswalk provides a clinical tool that can guide decisions around multidisciplinary care for geriatric trauma patients managed surgically and non-surgically.

## 5. Conclusions

Among geriatric trauma patients, a CCI value of zero excellently predicts an ASA-PS grade 1 and acceptably predicts an ASA-PS grade 2. CCI values of 1 and higher acceptably predict ASA-PS 3 or 4, traditionally described as the high-risk surgical group. While CCI values 2, 3, and 4 may predict ASA-PS grade 4, the higher the CCI value, the more the ASA-PS grade tends to be a grade 4. This study provides a crosswalk between CCI and ASA-PS if the CCI, as well as the age, sex, marital status, and body mass index, are known. 

## Figures and Tables

**Figure 1 healthcare-11-01137-f001:**
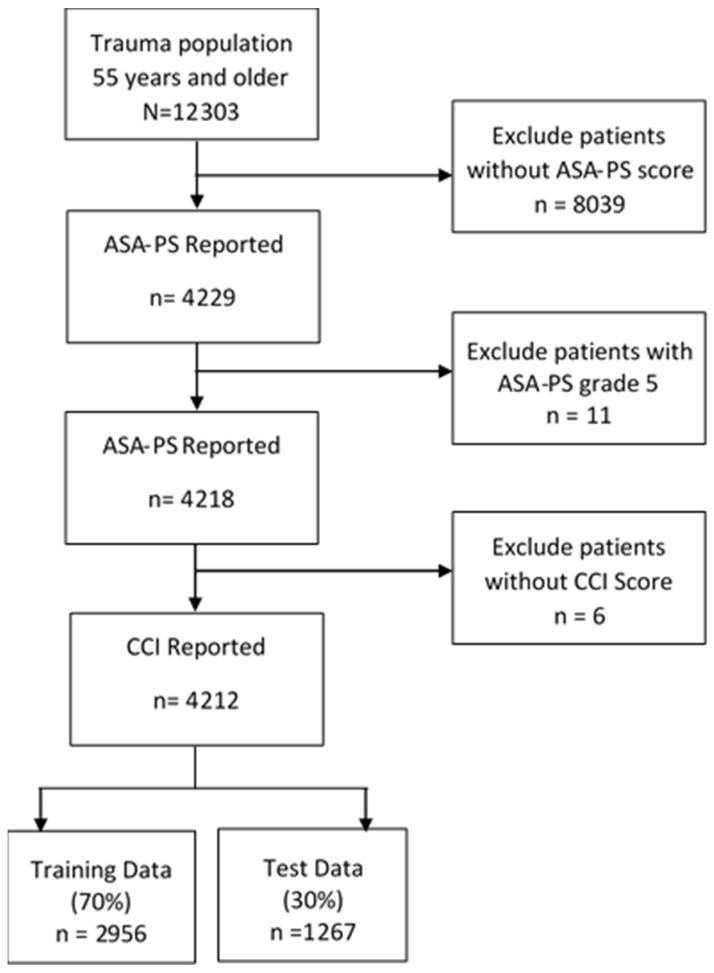
Data selection steps. ASA-PS: American Society of Anesthesiologists Physical Status; CCI: Charlson Comorbidity Index.

**Figure 2 healthcare-11-01137-f002:**
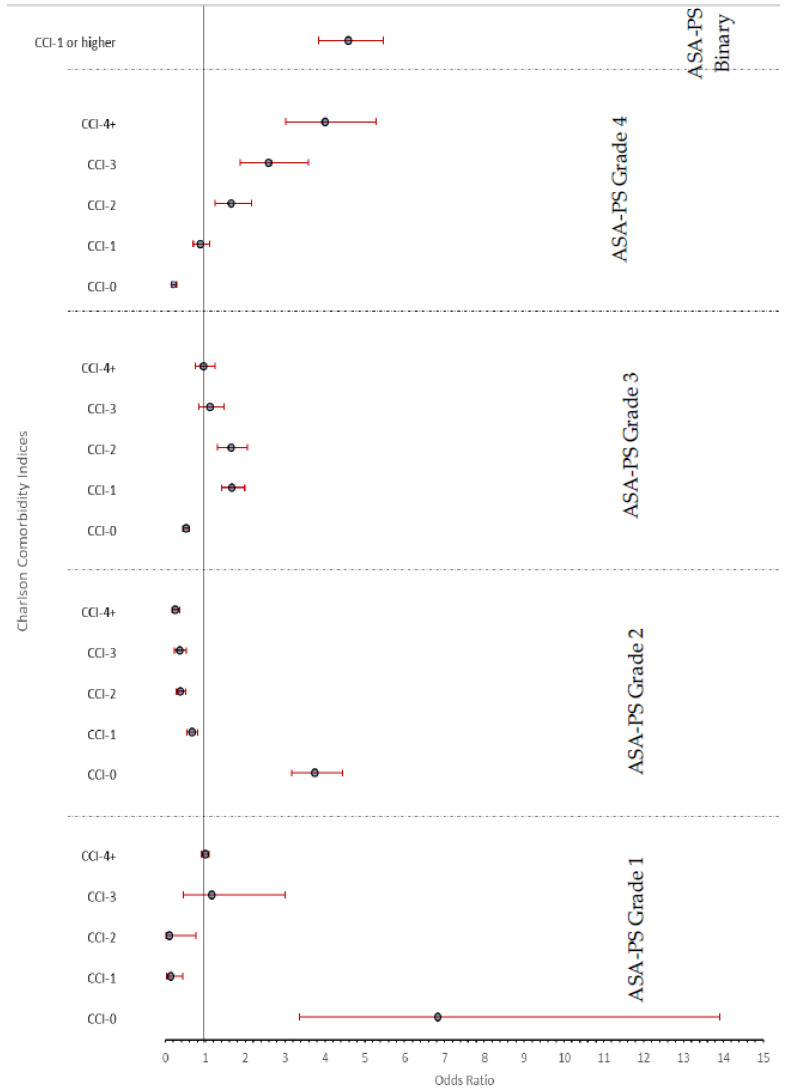
The likelihood of each Charlson Comorbidity Index predicting each of ASA-PS grades 1 to 4 among adults with geriatric trauma injuries using the training dataset. Each CCI index was a predictor dummy variable assessing each category of ASA-PS grades with each model selected using LASSO forward-selection tools.

**Figure 3 healthcare-11-01137-f003:**
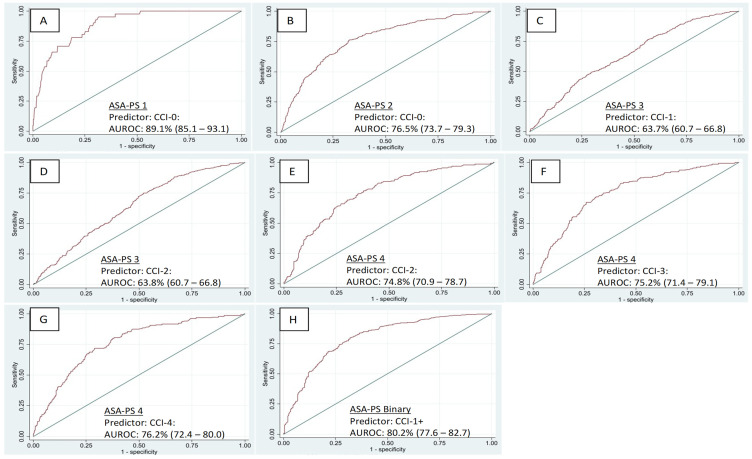
Area under the receiver operating characteristic curve assessing the accuracy of the most predictive CCI category in predicting each ASA-PS grade using the test dataset. (**A**) CCI-0 as a predictor of ASA-PS 1; (**B**) CCI-0 as a predictor of ASA-PS 2; (**C**) CCI-1 as a predictor of ASA-PS 3; (**D**) CCI-2 as a predictor of ASA-PS 3; (**E**) CCI-2 as a predictor of ASA-PS 4; (**F**) CCI-3 as a predictor of ASA-PS 4; (**G**) CCI-4 as a predictor of ASA-PS 4; (**H**) CCI-1+ as a predictor of ASA-PS Binary.

**Table 1 healthcare-11-01137-t001:** Demographic and injury characteristics of the study population in the training and test datasets.

Variables	Training Data (n = 2946) (70%)	Test Data (n = 1266) (30%)
Age (Mean (SD)) [years]	76.7 (12.7)	77.3 (12.8)
Sex		
Male	854 (29.0)	383 (30.0)
Female	2092 (71.0)	883 (70.0)
Marital Status		
Single	739 (25.1)	278 (21.9)
Married	1252 (42.5)	559 (44.2)
Divorced	300 (10.2)	134 (10.6)
Widowed	655 (22.2)	295 (23.3)
Body Mass Index		
Normal Body Mass Index	1220 (41.4)	524 (41.4)
Underweight	138 (4.7)	65 (5.1)
Overweight	1179 (40.0)	505 (39.9)
Obese	409 (13.9)	172 (13.6)
CCI (Categorical Measure)		
0	1284 (43.4)	554 (43.8)
1	769 (26.1)	324 (25.6)
2	391 (13.3)	183 (14.4)
3	226 (7.7)	92 (7.3)
4+	280 (9.5)	113 (8.9)
CCI (Continuous Measure)		
Mean (SD) *	1 (2)	1 (2)
Median (Q1, Q3)	1 (0, 2)	1 (0, 2)
ASA-PS Grade		
1	77 (2.6)	41 (3.2)
2	979 (33.2)	410 (32.4)
3	1473 (50.0)	656 (51.8)
4	417 (14.2)	159 (12.6)
ASA-PS Grade (Binary)		
1 and 2 (Low-risk)	1056 (35.8)	451 (35.6)
3 and 4 (High-risk)	1890 (64.2)	815 (64.4)

ASA-PS: American Society of Anesthesiologists Physical Status; CCI: Charlson Comorbidity Index; * Approximated to the nearest whole number.

**Table 2 healthcare-11-01137-t002:** Frequency distributions of the ASA-PS grade and the Charlson Comorbidity Index using the training dataset.

Variables	ASA-PS 1 (%)	ASA-PS 2 (%)	ASA-PS 3 (%)	ASA-PS 4 (%)	Total
CCI—0	68 (88.3)	659 (67.3)	496 (33.7)	57 (13.7)	1280 (43.4)
CCI—1	3 (3.9)	197 (20.1)	463 (31.4)	106 (25.4)	769 (26.1)
CCI—2	1 (1.3)	62 (6.3)	243 (16.5)	85 (20.4)	391 (13.3)
CCI—3	5 (6.5)	33 (3.4)	123 (8.3)	65 (15.6)	226 (7.7)
CCI—4+	0 (0.0)	28 (2.9)	148 (10.1)	104 (24.9)	280 (9.5)
Total	77 (100.0)	979 (100.0)	1473 (100.0)	417 (100.0)	2946 (100.0)

ASA-PS: American Society of Anesthesiologists Physical Status; CCI: Charlson Comorbidity Index.

**Table 3 healthcare-11-01137-t003:** Unadjusted odds of demographic characteristics and Charlson Comorbidity Index (CCI) predicting each ASA-PS grade category using the training dataset.

Variables	ASA-PS Grade 1 (95% CI)	ASA-PS Grade 2 (95% CI)	ASA-PS Grade 3 (95% CI)	ASA-PS Grade 4 (95% CI)	ASA-PS Grades 3 or 4 (Binary) (95% CI)
Age (in years)	**0.91 (0.89–0.94)**	**0.95 (0.94–0.96)**	**1.03 (1.02–1.04)**	**1.04 (1.03–1.05)**	**1.06 (1.05–1.07)**
Sex					
Male	Ref	Ref	Ref	Ref	Ref
Female	1.71 (0.97–3.02)	**1.33 (1.12–1.59)**	**0.84 (0.71–0.98)**	**0.79 (0.63–0.98)**	**0.72 (0.61–0.85)**
Marital Status					
Single	Ref	Ref	Ref	Ref	Ref
Married	1.00 (0.59–1.69)	**1.26 (1.05–1.53)**	0.85 (0.71–1.02)	0.88 (0.66–1.16)	**0.80 (0.66–0.96)**
Divorced	1.07 (0.50–2.28)	1.15 (0.87–1.52)	0.82 (0.63–1.07)	1.14 (0.77–1.70)	0.86 (0.66–1.14)
Widowed	**0.24 (0.09–0.63)**	**0.43 (0.34–0.56)**	**1.42 (1.14–1.75)**	**1.98 (1.49–2.64)**	**2.51 (1.97–3.21)**
Body Mass Index					
Normal Weight	Ref	Ref	Ref	Ref	Ref
Underweight	2.27 (0.97–5.31)	**0.67 (0.45–0.99)**	1.07 (0.75–1.52)	1.36 (0.85–2.17)	1.28 (0.88–1.88)
Overweight	1.46 (0.89–2.38)	1.05 (0.88–1.24)	0.87 (0.74–1.02)	1.13 (0.90–1.42)	0.92 (0.78–1.08)
Obese	0.31 (0.10–1.04)	0.86 (0.67–1.09)	1.24 (0.99–1.56)	0.96 (0.69–1.34)	1.25 (0.98–1.59)
CCI (Categorical Measure)					
0	Ref	Ref	Ref	Ref	Ref
1	**0.07 (0.02–0.22)**	**0.32 (0.27–0.39)**	**2.39 (1.99–2.87)**	**3.43 (2.45–4.80)**	**3.74 (3.08–4.55)**
2	**0.05 (0.01–0.33)**	**0.18 (0.13–0.24)**	**2.60 (2.06–3.28)**	**5.96 (4.17–8.53)**	**6.84 (5.11–9.16)**
3	0.40 (0.16–1.01)	**0.16 (0.11–0.24)**	**1.89 (1.42–2.51)**	**8.66 (5.85–12.82)**	**6.50 (4.51–9.38)**
4+	No Data	**0.10 (0.07–0.16)**	**1.77 (1.37–2.30)**	**12.68 (8.85–18.17)**	**11.83 (7.89–17.75)**
CCI (Continuous Measure)					
CCI	**0.35 (0.24–0.53)**	**0.55 (0.51–0.59)**	**1.12 (1.07–1.17)**	**1.52 (1.44–1.61)**	**1.94 (1.79–2.10)**

ASA-PS: American Society of Anesthesiologists Physical Status. Significant association in bold.

**Table 4 healthcare-11-01137-t004:** Summary of the discriminant properties of the most predictive CCI value and each ASA-PS Category.

Predicted ASA-PS Grade	Most Predictive CCI Category	Pseudo R^2^ (%)	Sensitivity	Specificity	Youden’s Index *	AUROC Training Dataset (%) **	DeLong’s Test (*p*-Value)
ASA-PS 1	CCI–0	20.50	88.8%	37.7%	61.9%	85.2 (80.4–90.0)	N/A
ASA-PS 2	CCI–0	15.80	68.6%	58.2%	43.0%	76.6 (74.7–88.4)	N/A
ASA-PS 3	CCI–1	4.36	51.1%	61.7%	22.1%	64.7 (62.7–66.7)	0.587 (CCI-2 vs. CCI-1)
ASA-PS 3	CCI–2	3.96	49.6%	62.6%	21.9%	64.4 (62.4–66.4)	
ASA-PS 4	CCI–2	6.65	60.2%	57.4%	28.1%	68.7 (66.0–71.3)	<0.001 (CCI-4 vs. CCI-2)
ASA-PS 4	CCI–3	7.42	60.0%	58.3%	29.9%	70.3 (67.7–72.9)	0.047 (CCI-4 vs. CCI-3)
ASA-PS 4	CCI–4+	9.71	60.7%	60.4%	34.9%	72.6 (70.1–75.2)	
ASA-PS Binary	CCI 1–4+ vs. CCI–0 (ref)	20.37	60.0%	69.3%	46.0%	79.5 (77.7–81.2)	N/A

* Youden’s index: ranges from 0 to 100% assuming sensitivity and specificity are of equal value; the higher the value, the better the discriminant property. ** AUROC values ranging from 0.7 to 0.8 are acceptable, 0.8 to 0.9 are excellent, and higher than 0.9 are outstanding.

**Table 5 healthcare-11-01137-t005:** Threshold and equations for computing predicted ASA-PS categories using multivariate ordinal logistic regression model on the full dataset.

Parameter	Variable	Regression Coefficient	Standard Error	*p*-Value	95% CI Lower	95% CI Upper
Threshold	ASA-PS Grade 1	−3.004	0.998	0.003	−4.960	−1.048
	ASA-PS Grade 2	−7.385	0.341	<0.001	−6.010	−4.761
	ASA-PS Grade 3	−13.265	0.456	<0.001	−12.784	−9.747
Predictors	CCI Categories					
		0.662	0.176	<0.001	0.318	1.006
	Age at Injury					
	Age	0.095	0.015	<0.001	0.066	0.124
	Sex					
	Male	Ref				
	Female	−0.908	0.304	0.003	−1.504	−0.312
	Marital Status					
	Single	Ref				
	Married	0.072	0.274	0.793	−0.465	0.609
	Divorced	−0.376	0.398	0.345	−1.157	0.405
	Widowed	0.306	0.530	0.563	−0.732	1.345
	BMI Categories					
	Normal Weight	Ref				
	Underweight	−0.997	0.459	0.030	−1.898	−0.097
	Overweight	0.173	0.263	0.510	−0.342	0.689
	Obese	1.841	0.618	0.003	0.631	3.051
Probability of ASA-PS 1	$=\frac{1}{(1+\exp\left(-\left(3.004-y\right))\right)}$					
Probability of ASA-PS 2	$=\frac{1}{(1+\exp\left(-\left(7.385-y\right))\right)}-\frac{1}{(1+\exp\left(-(3.004-y))\right)}$					
Probability of ASA-PS 3	$=\frac{1}{(1+\exp\left(-\left(13.265-y\right))\right)}-\frac{1}{(1+\exp\left(-(7.385-y))\right)}$					
Probability of ASA-PS 4	$=1-\frac{1}{(1+\exp\left(-\left(13.265-y\right))\right)}$					
y	$=0.662\times CCI+0.095x\left(Age\right)+\left(-0.908\left(if\;Female\right)\right)+0.072\left(if\;Married\right)+(-0.376\left(if\;Divorced)\right)+0.306\left(if\;Widowed\right)+(-0.997\left(if\;Underweight)\right)+0.173\left(if\;Overweight\right)+1.841(if\;Obese))$					

ASA-PS: American Society of Anesthesiologists Physical Status; CCI: Charlson Comorbidity Index; If statements in the equation imply multiplying by 1 if the condition attached to the specific if statement is satisfied, else multiply by 0.

## Data Availability

Deidentified data are available upon request.

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
