# Peer review of "Crosswalk between Charlson Comorbidity Index and the American Society of Anesthesiologists Physical Status Score for Geriatric Trauma Assessment"

_healthcare, 2023, doi:10.3390/healthcare11081137_

Round 1

Reviewer 1 Report

The authors sought to develop a linkage between CCI and ASA-PS by analyzing 4,223 adult trauma cases that were 55 years of age or older and had both ASA-PS and CCI values. They revealed that ASA-PS grades 1 or 2 were highly predicted by a CCI of zero, whereas ASA-PS grades 3 or 4 were highly predicted by a CCI of 1 or higher. Furthermore, while CCI of 3 accurately predicted ASA-PS grades 4, CCI of 4 and above demonstrated increased accuracy in predicting ASA-PS grades 4. I think the way research papers are currently formatted is well organized and suitable for publication in the journal Healthcare at MDPI. The manuscript might be approved for publication in this journal with only minor modifications. However, let me first provide some general and specific criticism of the authors.

General comments: The authors created a formula that may accurately situate a trauma patient in the appropriate ASA-PS grade after adjusting for age, sex, marital status, and body mass index. In my opinion, the current study style is appropriate for publication in MDPI's journal of healthcare. So, I think the work could be published in this journal with minor changes.

Specific comments:                        

  1. Title: The present title should be fixed.
  2. Abstract: The author should follow the journal guidelines for writing the abstract.
  3. Introduction: The author should rewrite the introduction with more references and can replace the table in the methodology section.
  4. Overall, the author did not adhere to the journal's guidelines for incorporating references throughout the manuscript.
  5. The author didn't write about how they calculated the odd ratio.
  6. In Figure 2, please separate the figure legend from the figure.
  7. Reference: The author didn't follow the instructions in the journal.
  8. The language and style of English must be extensively adjusted.

Author Response

Point-by-Point Response to Reviewer 1

The authors sought to develop a linkage between CCI and ASA-PS by analyzing 4,223 adult trauma cases that were 55 years of age or older and had both ASA-PS and CCI values. They revealed that ASA-PS grades 1 or 2 were highly predicted by a CCI of zero, whereas ASA-PS grades 3 or 4 were highly predicted by a CCI of 1 or higher. Furthermore, while CCI of 3 accurately predicted ASA-PS grades 4, CCI of 4 and above demonstrated increased accuracy in predicting ASA-PS grades 4. I think the way research papers are currently formatted is well organized and suitable for publication in the journal Healthcare at MDPI. The manuscript might be approved for publication in this journal with only minor modifications. However, let me first provide some general and specific criticism of the authors.

General comments:

The authors created a formula that may accurately situate a trauma patient in the appropriate ASA-PS grade after adjusting for age, sex, marital status, and body mass index. In my opinion, the current study style is appropriate for publication in MDPI's journal of healthcare. So, I think the work could be published in this journal with minor changes.

Response:

Thank you for your feedback.

Specific comments:

Comment                        

Title: The present title should be fixed.

Response

We have edited the title. The original version was “What About the Non-Surgical Geriatric Trauma Patient? Creating A Crosswalk Between the Charlson Comorbidity Index and the American Society of Anesthesiologists Physical Status Score”. The current title reads: “Crosswalk Between Charlson Comorbidity Index and the American Society of Anesthesiologists Physical Status Score for Geriatric Trauma Assessment.”

Comment

Abstract: The author should follow the journal guidelines for writing the abstract.

Response

We have edited the abstract. Specifically, we have removed the heading of the abstract subsections. The abstract is a single paragraph without subheadings. The abstract now reads:

“The American Society of Anesthesiologists Physical Status (ASA-PS) grade better risk stratifies geriatric trauma patients, but it is only reported in patients scheduled for surgery. Charlson Comorbidity Index (CCI), however, is available for all patients. This study aims to create a crosswalk from CCI to ASA-PS. Geriatric trauma cases, aged 55 years and older with both ASA-PS and CCI values (N= 4,223) were used for the analysis. We assessed the relationship between CCI and ASA-PS, adjusting for age, sex, marital status, and body mass index. We reported the predicted probabilities, and the receiver operating characteristics. A CCI of zero was highly predictive of ASA-PS grade 1 or 2 and a CCI of 1 or higher was highly predictive of ASA-PS grade 3 or 4. Additionally, while a CCI of 3 predicted ASA-PS grade 4, a CCI of 4 and higher exhibited greater accuracy in predicting ASA-PS grade 4. We created a formula that may accurately situate a geriatric trauma patient in the appropriate ASA-PS grade after adjusting for age, sex, marital status, and body mass index. In conclusion, ASA-PS grades can be predicted from CCI, and this may aid in generating more predictive trauma models.”

Comment

  1. Introduction: The author should rewrite the introduction with more references and can replace the table in the methodology section.

Response

We have rewritten the introduction. The specific changes we have made include (1) a new first paragraph that provides background on geriatric trauma, (2) more context on how CCI is defined, (3) a deleted Table 1 and we renumbered all the tables in the manuscript, and (4) an additional ten references to the previous 18 references.

The first paragraph of the Introduction reads:

“Geriatric trauma is the fifth leading cause of mortality among older adults in the U.S. and accounts for a quarter of inpatient geriatric admissions nationally [1]. These injuries commonly occur from falls, motor vehicle crashes, and burn injuries [2], and approximately 80 percent of geriatric trauma cases are managed non-surgically [3]. Geriatric trauma patients, aged 55 years and older [4], are a unique trauma population that is at increased mortality risk compared to trauma patients less than 55 years [5-9]. Multiple factors have been associated with increased geriatric trauma mortality risk some of which include frailty [10,11], cognitive impairment [1], cardiovascular and pulmonary insufficiency [1], and poor injury triage [6,12,13]. With the aging U.S. population [14], it is expected that geriatric trauma cases will increase over time. Improving injury outcomes among geriatric trauma patients, therefore, has been a priority area for quality assessment and improvement by the American College of Surgeons Trauma Quality Improvement Program [15].”

The additional information on the CCI in paragraph 3 of the Information reads:

“The CCI, a measure of patients’ mortality risks, assigns values ranging from one to six to distinct comorbidities [23,24]. These comorbid conditions include myocardial infarction, congestive cardiac failure, peripheral vascular disease, cerebrovascular accident or transient ischemic attack, dementia, chronic obstructive pulmonary disease, connective tissue disease, peptic ulcer disease, liver disease, diabetes, hemiplegia, chronic kidney disease, metastatic and non-metastatic solid tumor, leukemia, lymphoma, and acquired immune deficiency syndrome [23,24].”

Comment

  1. Overall, the author did not adhere to the journal's guidelines for incorporating references throughout the manuscript.

Response

We have corrected this error. Here is the summary of the changes we made:

  1. We have ensured that our in-text citations are before the punctuation marks.
  2. We ensured the appropriate abbreviation of all the Journal names in the reference list.

Comment

  1. The author didn't write about how they calculated the odd ratio.

Response

We have rewritten this section entirely to add some clarity to our analytical steps. The information on how the odds ratios were calculated and how the ROC analysis was performed are as follows:

“Using the ASA-PS binary models, we performed a Receiver Operating Characteristics (ROC) analysis to assess the discriminative property of CCI-based predictive probabilities on ASA-PS. Our ROC analysis was in two steps: (1) univariate logistic regression (2) computation of areas under of curve from predicted probabilities of the logistic regression analysis. For each set of univariate analysis, the five outcome measures were the four dummy definitions of the ASA-PS (for example ASA-PS 1: Yes or No, ASA-PS 2: Yes or No, etc.) and the high-risk/low-risk ASA-PS binary categorization. For each of these five binary outcome variables, the predictors were the five dummy variable definitions of CCI (for example CCI 0: Yes or No, CCI 1: Yes or No, etc.). Essentially, we had 25 univariate logistic regression models i.e. each of the five ASA-PS binary variables had five CCI dummy variable predictors. We performed the univariate logistic regression analyses using the training dataset and reported the unadjusted odds ratios for all 25 models and computed the predicted probabilities for each patient in the models that had significant associations. We validated the regression models by repeating the same regression analyses using the test dataset. We generated predicted probabilities for each patient in the models that showed significant associations. Of the 25 regression analyses, eight models showed significant associations between ASA-PS and CCI. Thereafter, we performed ROC analyses across the eight significant models using the training and test datasets. For each ROC analysis, the ASA-PS binary variable was the reference variable, and its associated predicted probability variable was the classification variable. We reported the pseudo-r-squared values, the sensitivity, specificity, and the area under the curve of each ROC analysis (AUROC). We interpreted the AUROC ranging from 0.7 to 0.8 as acceptable, 0.8 to 0.9 as excellent, and values higher than 0.9 as outstanding [43]. Also, we computed the Youden index, a measure of the model’s effectiveness as a diagnostic marker [44]. Values of 0.5 or higher were deemed adequate [44]. For ASA-PS models that had more than one significant CCI predictor, we assessed the differences in the AUROC using DeLong's test [45].

See Methods: Model Testing and Internal Validation

Comment:

  1. In Figure 2, please separate the figure legend from the figure.

Response

We have made this edit. The figure legend is separate from the figure.

Comment

  1. Reference: The author didn't follow the instructions in the journal.

Response

We have corrected all errors in the reference list. Thank you.

Comment

  1. The language and style of English must be extensively adjusted.

Response

We have edited the language of the manuscript, corrected the grammar, and ensured correct spelling.

Reviewer 2 Report

In this study, the authors developed a crosswalk formula for estimating ASA-PS grades from CCI. The sample size and statistical methods are adequate, the paper is well written. The results might be useful for researchers and practicioners not only in trauma medicine, but also health services research and related areas. 

I have some minor comments: 

- Please provide some more detail on covariate selection. For instance, why was marital status, but not smoking status included? 

- Please expand on the limitations: Single center study, analysis restricted on ages 55+, thus limited generalizibility

- Please discuss practical implications and target audience: research / quality assessment vs. clinical practice. 

Author Response

Point-by-Point Response to Reviewer 2

General Comment

In this study, the authors developed a crosswalk formula for estimating ASA-PS grades from CCI. The sample size and statistical methods are adequate, the paper is well written. The results might be useful for researchers and practitioners not only in trauma medicine, but also health services research and related areas. 

Response

Thank you for your feedback

I have some minor comments: 

Comment

- Please provide some more detail on covariate selection. For instance, why was marital status, but not smoking status included? 

Response

We have clarified how the covariates were selected.  The sentences read:

“Our first step in creating a crosswalk between ASA-PS and CCI was to select the variables in the model. The main predictor was CCI, measured as a categorical variable. We selected covariates that are conventional demographic characteristics, similar to prior studies [22,33]. We limited our covariate selection to variables that are routinely entered in the electronic health records (such as age, sex, marital status, and body mass index) and excluded variables that are typically associated with high rates of under-reporting (such as smoking history) or non-response (such as race/ethnicity).”

See Methods: ASA-PS and CCI Crosswalk

Comment

- Please expand on the limitations: Single center study, analysis restricted on ages 55+, thus limited generalizability

Response

Thank you. We have added this limitation. The addition reads:

“Also, we pooled data from a single center and our data is limited to adults 55 years and older. Hence, our study has limited generalizability.”

Discussion: Paragraph 5

Comment

- Please discuss practical implications and target audience: research / quality assessment vs. clinical practice. 

Response

We have added a paragraph in the discussion that addresses the practical implications of the crosswalk for researchers and clinicians. The additional sentences read:

“This crosswalk provides a more objective measure of ASA-PS allocation for pre-operative care, and early assignment of ASA-PS grade may inform the need for geriatric consult activation or multi-disciplinary specialty care. Additionally, the crosswalk provides an opportunity for researchers to extend the frontiers of geriatric trauma care by creating predictive tools that will improve injury triage.”

See Discussion: Paragraph 4

Reviewer 3 Report

Dear authors

Thank you for submitting your article in journal. Your research article is very interesting because the both of Charlson Comorbidity Index (CCI) and American Society of Anesthesiologists Physical Status Score (ASA-PS) are widely use in clinical practice for surgical and trauma patients. However, the introduction and discussion need some revision and more descriptive about the distinctive point and limitation of each scoring system. I have some comments on your manuscript.

Major comment

1.     Abstract: Because the target patients which plan to crosswalk between the CCI and ASA-PS is geriatric people. So, the abstract should add the “geriatric” and specific description for geriatric patients.

2.     Introduction: Because the target patients which plan to crosswalk between the CCI and ASA-PS is geriatric people. The definition of geriatric and middle -aged should be described in the aim of study under main manuscripts.

3.     Materials and Methods > 2.1 Study Design, Setting, and Patients: The statement “. This study was part of the validation of studies for the implementation of a novel score for trauma triage in Geriatric and Middle-Aged patients (STTGMA).“

-      For your manuscript, “Dose the middle-aged patients were included in your study ?” Please describe in the manuscript with the reasons “Why or Why not included the middle-aged or specified only the geriatric group ?”

4.     Abstract > Conclusion: the statements “ASA-PS grades can be computed from CCI, and this may aid in generating more predictive trauma models.” 

-      The predictability and implication to practice of ASA-PS and CCI scoring system were developed via the validating system in a different group of patients. So, please avoid the word “computed” in the conclusion. The terms “predict”, “apply” or “refer” were recommended.

5.     The benefit of a crosswalk between ASA-PS and CCI should be more described in the introduction part.

6.     The comparison between CCI and ASA-PS should be more described in the Discussion part.

-      Example: Pro and Con of CCI and ASA-PS / Specific group of trauma patients (in the other aspects [not only for surgery and non-surgery group])

Minor comments 

1.     Materials and Method: the statement “Data were analyzed with STATA version 17” The full description of the STATA program was required for a manuscript which plans for publication. 

a.     Example “The statistical analyses were performed with STATA/SE 16.0 for Mac (StataCorp, TX, USA)”

2.     Figure 2 which demonstrates the likelihood of each CCI predicting each ASA-PS Grades 1 to 4 were not identified the grade number of ASA-PS. 

a.     Each category of ASA-PS grades should be identified the grade number in each row in Y-axis [Example ASA-PS Grade 1, 2 ,3,….] 

3.     Figure 3: The numerical data of the Area under the ROC curve should be described in all figures (A-H).

4.     The references should revise to the same format. Example Full page or Abbreviate page numbers.

Best wishes

Reviewer  

Author Response

Point-by-Point Response to Reviewer 3

Comment

Dear authors

Thank you for submitting your article in journal. Your research article is very interesting because the both of Charlson Comorbidity Index (CCI) and American Society of Anesthesiologists Physical Status Score (ASA-PS) are widely use in clinical practice for surgical and trauma patients. However, the introduction and discussion need some revision and more descriptive about the distinctive point and limitation of each scoring system.

Response

Thank you for your feedback. We have made substantial changes to our introduction and discussion. We have added more descriptives to elaborate on the distinctive point and limitations of the ASA-PS and CCI.

Major comment

I have some comments on your manuscript.

Comment

Abstract: Because the target patients which plan to crosswalk between the CCI and ASA-PS is geriatric people. So, the abstract should add the “geriatric” and specific description for geriatric patients.

Response

Thank you for your comment. We have added the term “geriatric” and our specific descriptor of the term “geriatric” in the abstract. The abstract now reads:

“The American Society of Anesthesiologists Physical Status (ASA-PS) grade better risk stratifies geriatric trauma patients, but it is only reported in patients scheduled for surgery. Charlson Comorbidity Index (CCI), however, is available for all patients. This study aims to create a crosswalk from CCI to ASA-PS. Geriatric trauma cases, aged 55 years and older with both ASA-PS and CCI values (N= 4,223) were used for the analysis. We assessed the relationship between CCI and ASA-PS, adjusting for age, sex, marital status, and body mass index. We reported the predicted probabilities, and the receiver operating characteristics. A CCI of zero was highly predictive of ASA-PS grade 1 or 2 and a CCI of 1 or higher was highly predictive of ASA-PS grade 3 or 4. Additionally, while a CCI of 3 predicted ASA-PS grade 4, a CCI of 4 and higher exhibited greater accuracy in predicting ASA-PS grade 4. We created a formula that may accurately situate a geriatric trauma patient in the appropriate ASA-PS grade after adjusting for age, sex, marital status, and body mass index. In conclusion, ASA-PS grades can be predicted from CCI, and this may aid in generating more predictive trauma models.“

See Abstract

Comment

Introduction: Because the target patients which plan to crosswalk between the CCI and ASA-PS is geriatric people. The definition of geriatric and middle -aged should be described in the aim of study under main manuscripts.

Response 

Thank you for this comment. We have defined “geriatric trauma patients” at the point the phrase was first mentioned in the first paragraph of the introduction. The introduction (newly added based on comments from Reviewer 1) reads:

“Geriatric trauma is the fifth leading cause of mortality among older adults in the U.S. and accounts for a quarter of inpatient geriatric admissions nationally [1]. These injuries commonly occur from falls, motor vehicle crashes, and burn injuries [2], and approximately 80 percent of geriatric trauma cases are managed non-surgically [3]. Geriatric trauma patients, aged 55 years and older [4], are a unique trauma population that is at increased mortality risk compared to trauma patients less than 55 years [5-9]. Multiple factors have been associated with increased geriatric trauma mortality risk some of which include frailty [10,11], cognitive impairment [1], cardiovascular and pulmonary insufficiency [1], and poor injury triage [6,12,13]. With the aging U.S. population [14], it is expected that geriatric trauma cases will increase over time. Improving injury outcomes among geriatric trauma patients, therefore, has been a priority area for quality assessment and improvement by the American College of Surgeons Trauma Quality Improvement Program [15].”

See Introduction: Paragraph 1

The aim and hypothesis of the study retained the term “geriatric”, as follows:

“The aim of this study, therefore, is to create a crosswalk between CCI and ASA-PS by predicting ASA-PS grades using CCI values while evaluating the accuracy of such predictions among geriatric trauma patients. We hypothesized that the CCI will exhibit acceptable levels of accuracy in predicting ASA-PS grades among geriatric trauma patients.” 

See Introduction: Paragraph 4

Comment

Materials and Methods > 2.1 Study Design, Setting, and Patients: The statement “. This study was part of the validation of studies for the implementation of a novel score for trauma triage in Geriatric and Middle-Aged patients (STTGMA).“

-      For your manuscript, “Dose the middle-aged patients were included in your study ?” Please describe in the manuscript with the reasons “Why or Why not included the middle-aged or specified only the geriatric group ?”

Response

We have edited this section. This study was one of the validation studies aimed at developing an injury risk triage tool for geriatric trauma patients. However, this study simply focused on a measure of creating a crosswalk that will be of benefit to our future studies.

Essentially, this study and the larger study protocol were limited to persons aged 55 years and older. When the STTGMA was originally modeled in 2018, persons aged 55 to 64 we classified as middle age. The seminal STTGMA paper in 2018 did not include persons typically classified as middle-aged adults i.e 45 to 64 years but was restricted to patients 55 years and older. However, more recent studies have reported that injury-related mortality significantly increases from age 55 years and older (see Fakhry et al., 2021 (https://doi.org/10.1097/TA.0000000000003062) and Konda et al., 2020 (https://doi.org/10.1177/2151459320955087 ). For this study, we defined the geriatric population as patients aged 55 years and older and we added a rationale for this decision, for clarity.

The changes we made are as follows:

“For this retrospective cohort study, we pooled a single institutional trauma data between 2014 and 2020 from the electronic health record of an urban academic hospital with level-I trauma center designation in New York. This study was part of the validation studies for the implementation of a novel scoring tool for geriatric trauma triage (Institutional Review Board (IRB): s15-00371) and the protocol of the investigation has been published earlier [30-33]. All relevant de-identified patient data were recorded in an IRB-approved database

See Materials and Methods:  Study Design, Setting, and Patients

“The inclusion criteria consisted of adult patients 55 years and older, admitted through the Emergency Department, and who received Orthopedic trauma care between 2014 and 2020 (N = 12,303) (Figure 1). We used age 55 as the cut-off since injury-related mortality significantly increases from age 55 years and higher [4].

See Materials and Methods:  Inclusion and Exclusion criteria

Comment

 Abstract > Conclusion: the statements “ASA-PS grades can be computed from CCI, and this may aid in generating more predictive trauma models.” The predictability and implication to practice of ASA-PS and CCI scoring system were developed via the validating system in a different group of patients. So, please avoid the word “computed” in the conclusion. The terms “predict”, “apply” or “refer” were recommended.

Response

We have made this edit. The conclusion of the abstract now reads:

“In conclusion, ASA-PS grades can be predicted from CCI, and this may aid in generating more predictive trauma models.

See Abstract: Conclusion

Comment

The benefit of a crosswalk between ASA-PS and CCI should be more described in the introduction part.

Response:

We have added an additional description of the benefit of the crosswalk. The added sentence read:

“Providing a crosswalk that will identify a predicted ASA grade of all geriatric trauma patients will provide a uniform measure of clinical assessment without excluding the larger proportion of geriatric trauma patients that are managed non-surgically.”

See Introduction: Paragraph 4

Comment

The comparison between CCI and ASA-PS should be more described in the Discussion part.

-      Example: Pro and Con of CCI and ASA-PS / Specific group of trauma patients (in the other aspects [not only for surgery and non-surgery group])

Response

We have described the comparison between CCI and ASA-PS in the discussion. The added paragraph is as follows:

“Indeed, neither the CCI nor the ASA-PS is without weaknesses, but this crosswalk provides an opportunity for continued research to improve geriatric trauma outcomes. The strengths of the CCI, for example, include its relative ease of calculation, and its availability for both surgical and non-surgical geriatric trauma patients. However, the CCI is limited to 16 chronic conditions [24], and since its development in 1987, the weights allocated to each illness category has become obsolete due to advancement in clinical care [26]. Additionally, three decades after its creation, more chronic conditions have been identified as significant predictors of mortality among older adults some of which include such as valvular heart disease, cardiac arrhythmias, and alcohol and drug abuse [26,51]. We circumvented this limitation by calculating the index of diseases and using age as a separate predictor, consistent with Mannion et al.'s seminal paper [22]. The ASA-PS, while being a more predictive measure of morbidity and mortality [28,29], is limited by being limited to patients scheduled for surgery, officially assigned by an anesthesiologist on the day of the surgery [52], and its weak inter-rater agreement [22,53]. This crosswalk provides a more objective measure of ASA-PS allocation for pre-operative care, and early assignment of ASA-PS grade may inform the need for geriatric consult activation or multi-disciplinary specialty care. Additionally, the crosswalk provides an opportunity for researchers to extend the frontiers of geriatric trauma care by creating predictive tools that will improve injury triage.”

See Discussion: Paragraph 3

Minor comments 

 Comment

  1. Materials and Method: the statement “Data were analyzed with STATA version 17” The full description of the STATA program was required for a manuscript which plans for publication. 
  2. Example “The statistical analyses were performed with STATA/SE 16.0 for Mac (StataCorp, TX, USA)”

Response

We have made this edit. The sentence now reads:

“Data were analyzed with STATA version 17 for Windows (StataCorp, TX, USA) [48].”

See Materials and Method

Comment

  1. Figure 2 which demonstrates the likelihood of each CCI predicting each ASA-PS Grades 1 to 4 were not identified the grade number of ASA-PS. 
  2. Each category of ASA-PS grades should be identified the grade number in each row in Y-axis [Example ASA-PS Grade 1, 2 ,3,….] 

Response

We have made this correction. Each category of ASA-PS grade is identified on the y-axis.

Comment

  1. Figure 3: The numerical data of the Area under the ROC curve should be described in all figures (A-H).

Response

We have added the numerical data of the area under the curve to each figure.

Comment

  1. The references should revise to the same format. Example Full page or Abbreviate page numbers.

Response

 We have changed all journal names to abbreviated journal names consistent with the journal requirement.